# The mental health and wellbeing of spouses, partners and children of emergency responders: A systematic review

Marie-Louise Sharp[1]*, Noa Solomon[2], Virginia Harrison[3,4], Rachael Gribble[1], Heidi Cramm[5], Graham Pike[3,4], Nicola T. Fear[1,6]

**1** Department of Psychological Medicine, King's Centre for Military Health Research, King's College London, London, United Kingdom, **2** Care Quality Commission, London, United Kingdom, **3** Faculty of Arts and Social Sciences, School of Psychology and Counselling, The Open University, London, United Kingdom, **4** The Centre for Policing Research and Learning, The Open University, London, United Kingdom, **5** Faculty of Health Sciences, School of Rehabilitation Therapy, Queen's University, Kingston, Canada, **6** Academic Department of Military Mental Health, Department of Psychological Medicine, King's College London, London, United Kingdom

* marie-louise.sharp@kcl.ac.uk

**Data Availability Statement:** All data are available by accessing original journal articles included in the systematic review. Additionally Supporting

## Abstract

Emergency responders (ERs), often termed First Responders, such as police, fire and paramedic roles are exposed to occupational stressors including high workload, and exposure to trauma from critical incidents, both of which can affect their mental health and wellbeing. Little is known about the impact of the ER occupation on the mental health and wellbeing of their families. The aim of the current study was to investigate what mental health and wellbeing outcomes and experiences have been researched internationally in ER families, and to examine the prevalence and associated risk and protective factors of these outcomes. We conducted a systematic review in accordance with an *a priori* PROSPERO approved protocol (PROSPERO 2019 CRD42019134974). Forty-three studies were identified for inclusion. The majority of studies used a quantitative, cross-sectional design and were conducted in the United States; just over half assessed police/law enforcement families. Themes of topics investigated included: 1) Spousal/partner mental health and wellbeing; 2) Couple relationships; 3) Child mental health and wellbeing; 4) Family support and coping strategies; and 5) Positive outcomes. The review identified limited evidence regarding the prevalence of mental health and wellbeing outcomes. Family experiences and risk factors described were ER work-stress spillover negatively impacting spousal/partner wellbeing, couple relationships, and domestic violence. Traumatic exposure risk factors included concerns family had for the safety of their ER partner, the negative impact of an ER partners' mental health problem on the couples' communication and on family mental health outcomes. Protective factors included social support; however, a lack of organisational support for families was reported in some studies. Study limitations and future research needs are discussed. Progressing this area of research is important to improve knowledge of baseline needs of ER families to be able to target interventions, improve public health, and support ER's operational effectiveness.

Information S1 Table summarises in detail underlying data from all papers.

**Funding:** NF is part funded by a UK Ministry of Defence grant, is a trustee of Help for Heroes - a charity supporting the wellbeing of veterans and their families, and a member of the Royal Foundation Emergency Responder's Senior Leader Board.

**Competing interests:** NF is part funded by a UK Ministry of Defence grant, is a trustee of The Warrior Programme - a charity supporting the wellbeing of veterans, service personnel and their families, a specialist academic member of the Independent Group Advising UK NHS Digital on the Release of Patient Data and Chair of the Royal Foundation Emergency Responder Senior Leader Board. This does not alter our adherence to PLOS ONE policies on sharing data and materials.

## Introduction

International research finds those in Emergency Responder (ER)/First Responder roles such as police, fire and ambulance personnel experience higher levels of depression, alcohol misuse and posttraumatic stress disorder (PTSD) compared to the general population due to increased exposure to trauma and organisational stressors [1–3]. There is less known about the concurrent effects of ER occupations on the mental health and wellbeing of ER families, but these families potentially experience secondary trauma and stresses that cascade over from their ER partner's work. The spillover stress model describes how the stressful experiences of one partner can impact on the psychosocial outcomes of their partners and impact family life [4,5]. Research conducted with families in occupational contexts with comparable trauma exposure and work-related stressors, including military families, indicates they have poorer family mental health, wellbeing and functioning compared to general population families due to experiences of absence, lone parenting, concern for their spouse/partner safety, and secondary trauma [6–8]. It is possible that ER families experience secondary traumatic stress (STS), which mirrors symptoms of PTSD, as it can be experienced by those who have intimate relationships with individuals who experience first-hand trauma [9].

Wellbeing is a critical factor to investigate in ER family's health as reviews have found subjective wellbeing to be influential in physical and mental health as well as longevity [10,11]. Wellbeing has been measured in large scale national studies to encompass areas of: satisfaction with life, the extent to which individual's feel the things they do in life are worthwhile, and how happy or anxious individuals report feeling [12]. Wellbeing has also been defined to encompass positive relations with others, autonomy, personal growth [13], social support and relationships [14] and resilience [15].

To our knowledge there is currently no systematic review that brings together international evidence regarding the mental health and wellbeing of ER families. Due to ER family's potential exposures to ER occupational stresses and secondary trauma, the objectives of this systematic review are to assess:

1. What types of mental health and wellbeing outcomes have been investigated in ER families, including those of spouses/partners and children?

2. What is the prevalence of positive and negative mental health and wellbeing outcomes in ER spouses/partners and children?

3. What are the experiences of ER spouses/partners and children with regards to mental health and wellbeing?

4. What are the risk and protective factors that are associated with an ER family's mental health and wellbeing?

The rationale for this systematic review is that ER family's mental health and wellbeing outcomes are pertinent for public health services in preventing ill health and supporting family and community health. Investigating these outcomes is also crucial for ER employers with regard to planning wellbeing support, supporting investment returns in recruitment and retention, and maintaining operational effectiveness. This review was conducted during the Covid-19 pandemic and therefore its findings are also relevant in light of additional pressures faced by ERs and their families, and understanding what support might be needed in the future [16,17].

## Materials and methods

### Inclusion/Exclusion criteria

For the purposes of the review, we include those families who had an ER member involved in traditional emergency services or volunteer roles–for example, those on an emergency call or at the scene of an emergency (police, fire, ambulance, air ambulance, mountain rescue, coast guard) and those in public safety roles who by nature of their position may be exposed to trauma and high stress situations (emergency call handlers, those viewing objectionable materials as part of investigations into terrorism, or child exploitation). Our definition of ERs and their families was purposively expansive to reflect broader definitions of modern family structures as advised by Gribble, Mahar [18] to include spouses/civil partners, cohabiting or intimate partners, parents of, and children/dependents within the ER community.

The inclusion criteria included empirical studies, where full text articles could be accessed, both quantitative and qualitative, published in peer-reviewed journals from the year 2000 onwards that assessed mental health and wellbeing in families of ER populations. Exclusion criteria included studies examining physical health in ER families or studies assessing families of general emergency department roles (such as nurses or doctors) with no pre-hospital emergency responsibilities. For a detailed list of all definitions and inclusion/exclusion criteria please see supporting information in S1 File.

### Search strategy

The literature search was conducted in May 2021 in accordance with an *a priori* PROSPERO approved protocol (PROSPERO 2019 CRD42019134974). Relevant studies published from the year 2000 onwards in peer-reviewed journals were identified through electronic searches on the following databases: MEDLINE, PsycINFO, Embase, Web of Science, PILOTS, EBSCO and CINAHL. Key search terms related to 'mental health', 'wellbeing', 'emergency responders' and 'family' were combined with appropriate Boolean operators. Please see supporting information S2 File. for full search terms used.

All returns were imported into 'Covidence', a systematic review programme which automatically identified duplicates and removed them upon importation [19]. Titles and abstracts were reviewed (N.S.), with ten per cent reviewed by a second author (M.L.S) to ensure the eligibility and exclusion criteria were consistently applied. After eligibility and exclusion criteria were applied, full texts were reviewed further for eligibility (N.S and M.L.S) and assessed as to whether they contained extractable primary data. Where researchers differed in their decisions to include or exclude papers, divergences were discussed, and a consensus was reached. The reference lists of all eligible studies were checked for any additional papers and eligible studies were added to Covidence for extraction. External stakeholders such as Emergency Responder organisations and charities and authors (H.C, N.F, G.P and G.H) were also asked to view the final reference list and indicate any additional studies they deemed eligible for inclusion. The inclusion of external stakeholders was to ensure effective patient and public involvement.

### Data extraction and analysis

Data extraction was conducted by two of the authors (M.L.S and N.S). Data from 43 papers were extracted, which included information on author, title and date of publication, responder population (i.e., firefighters, police officers, ambulance/paramedics), family relationship to the responder (e.g., spouse, child), overall sample size, sub-sample sizes (e.g. no. of firefighters/police/paramedics), study design, data collection method, response rate, mental health/wellbeing topic area (e.g. impact of PTSD on couple relationships), mental health/wellbeing measures

(including references of instruments used, amount of items, Likert scale treatment (e.g. summing or averaging score measures), internal reliability assessed via Cronbach's alpha scores, and key variables measured) (supporting information, S1 Table).

Data were also extracted on the means, standard deviations, and prevalence of mental health items. The methods and reporting of the results of this systematic review are described according to PRISMA (Preferred Reporting Items for Systematic Reviews and Meta-Analyses [20]). Extracted quantitative and qualitative data were synthesised using a narrative synthesis approach detailed in Popay, Roberts [21]. A preliminary synthesis was created by grouping and clustering data. These data were further translated utilising reflexive thematic analysis as detailed in Braun and Clarke [22] to collate data into themes and concepts, identifying where there was conceptual overlap. Data were then assessed to explore relationships within and between studies utilising conceptual mapping [23]. This final synthesis of data was tested with all review authors and shared with key stakeholders working operationally within ER organisations to ensure interpretation of results reflected lived and operational experience. Finally, a quality analysis was conducted to assess the robustness of data included in the synthesis to inform the systematic review's strengths and limitations.

## Quality analysis

To critically evaluate the articles, a quality appraisal tool was adapted from the Consolidated Criteria for Reporting Qualitative Research (COREQ) [24] and The Quality Assessment Tool for Observational Cohort and Cross-Sectional Studies (NIH Quality Assessment Tools [25]). The quality of the eligible papers was assessed utilising six different criteria for qualitative and quantitative studies. A score of '1' was awarded for each criterion that was met, and '0' for each that was not. Scores were then summed (and an average calculated for mixed methods papers) and studies classified as either: (a) poor quality (score 0–2); (b) fair (score 3–4); or (c) good (score 5–6) (for a full description of the quality analysis tool please see supporting information –S3 File). One researcher (N.S) scored the studies and 10% were reviewed by a second researcher (M.L.S) to ensure consistent application of quality scores.

## Results and discussion

### Study selection

Searches returned 1098 articles that met the initial criteria (Fig 1). Of these, 928 abstracts were excluded as they did not meet the full inclusion criteria. Of the 170 papers that remained, 131 papers were removed after full text review, leaving 39 articles for full data extraction. After cross-referencing for eligible papers and sharing the list with the other authors, four more articles were identified for inclusion, resulting in a total of 43 papers.

### Overview of studies

Of the 43 included papers, 27 were based on quantitative data, 13 on qualitative data and three were mixed method studies (Table 1). The majority of papers (n = 39) were cross-sectional studies, with four studies using two time points or longitudinal study designs. There were 24 papers that used police/law enforcement family samples, eight papers mixed samples of ER families, seven fire service families, and four papers ambulance/paramedics/emergency medicine practitioners' families. The majority of papers were based on study samples from the United States (n = 33), with the remainder from Australia (n = 5), Canada (n = 3), Israel (n = 1), and South Africa (n = 1). Twenty-two studies collected data directly from the spouse/partner of the ERs, six studies collected data directly from children of ERs

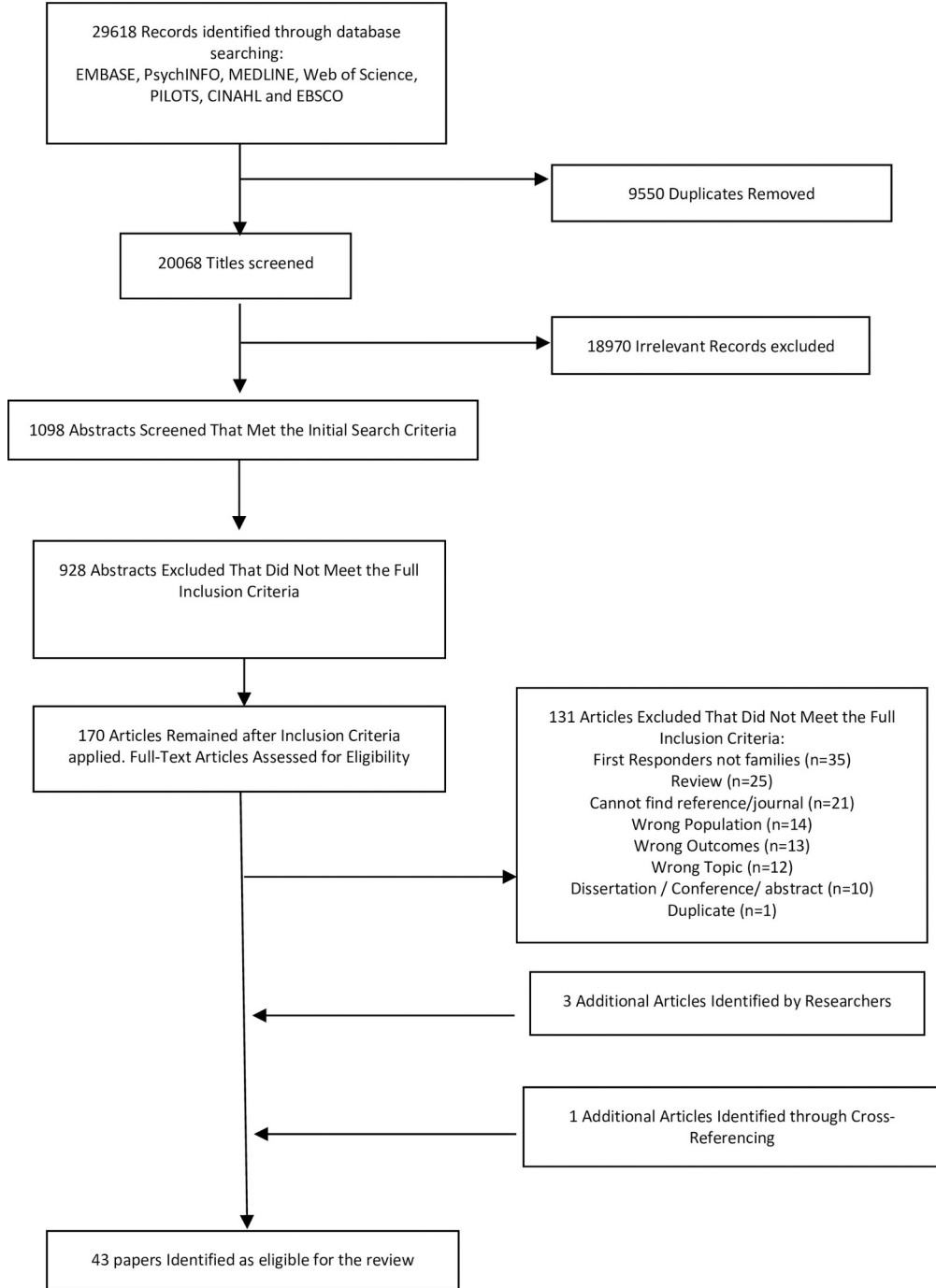

**Fig 1. PRISMA flow diagram study selection emergency responder families research.**

and 17 studies collected data only from the ER partner. Themes identified from the synthesis included 1) Spousal/partner mental health and wellbeing; 2) Couple relationships; 3) Child mental health and wellbeing; 4) Family support and coping strategies; and 5) Positive outcomes. 18 papers were assessed as good quality, 24 papers fair quality and 1 paper poor quality.

**Table 1. Study characteristics overview.**

| | |
|---|---|
| **Country** | **Australia (n = 5)[a]**: Davidson 2006 [35], McKeon 2020 [38]; Shakespeare-Finch 2002 [66]; Thompson 2005 [50]; Waddell 2020 [51]<br>**Canada (n = 3)**: King 2014 [42]; Regehr 2005 [27] (a); Regehr 2005 [27] (b)<br>**Israel (n = 1):** Kishon 2020 [64]<br>**South Africa (n = 1):** Wheater 2017 [34]<br>**United States (n = 33)**: Anderson 2011 [54]; Bochantin 2016 [44]; Brady 2019 [43]; Brimhall 2016 [41]; Brodie 2012 [29]; Comer 2014 [60]; Craun 2015 [46]; Duarte 2006 [61]; Erwin 2005 [59]; Gibson 2001 [53]; Haddock 2016 [40]; Halbesleben 2010 [67]; Helfers 2020 [65]; Hoven 2009 [62]; Johnson 2005 [58]; Karaffa 2015 [30]; Landers 2020 [32]; Meffert 2014 [36]; Menendez 2006 [45]; McCoy 2009 [39]; Pfefferbaum 2002 [68]; Porter 2016 [31]; Richardson 2016 [69]; Roberts 2001 [47]; Roberts 2013 [49]; Roth 2009 [28]; Ryan 2000 [52]; Sanford 2017 [48]; Uchida 2018 [63]; Watkins 2021 [33]; Zavala 2013 [55]; Zavala 2015 [56]; Zavala 2019 [57] |
| **Police Family Study (n = 24)** | Anderson 2011 [54]; Brady 2019 [43]; Brimhall 2016 [41]; Brodie 2012 [29]; Comer 2014 [60]; Craun 2015 [46]; Davidson 2006 [36]; Erwin 2005 [59]; Gibson 2001 [53]; Halbesleben 2010 [67]; Helfers 2020 [65]; Johnson 2005 [58]; Karaffa 2015 [30]; Landers 2020 [32]; Meffert 2014 [36]; McCoy 2009 [39]; Roberts 2001 [47]; Roberts 2013 [49]; Ryan 2000 [52]; Thompson 2005 [50]; Uchida 2018 [63]; Zavala 2013 [56]; Zavala 2015 [56]; Zavala 2019 [57] |
| **Firefighter Family Study (n = 7)** | Haddock 2016 [40]; Menendez 2006 [45]; Pfefferbaum 2002 [68]; Regehr 2005 [27] (b) Richardson 2016 [69]; Sanford 2017 [48]; Watkins 2021 [33] |
| **Ambulance/Paramedic Family Study (n = 4)** | King 2014 [42]; Shakespeare-Finch 2002 [66]; Regehr 2005[27] (a); Roth 2009 [28] |
| **Mixed ER Family Study (n = 8)** | Bochantin 2016 [44]; Duarte 2006 [61]; Hoven 2009 [62]; Kishon 2020 [64]; McKeon 2020 [38]; Porter 2016 [31]; Waddell 2020 [51]; Wheater 2017 [34] (8) |
| **Quantitative Study (n = 27)** | Anderson 2011 [54]; Brimhall 2016 [41]; Comer 2014 [60]; Craun 2015 [46]; Davidson 2006 [36]; Duarte 2006 [61]; Erwin 2005 [59]; Gibson 2001 [53]; Haddock 2016 [40]; Halbesleben 2010 [67]; Johnson 2005 [58]; King 2014 [42]; Kishon 2020 [64], McKeon 2020 [38]; Meffert 2014 [36]; McCoy 2009 [39]; Pfefferbaum 2002 [68]; Roberts 2001 [47]; Roberts 2013 [49]; Ryan 2000 [52]; Sanford 2017 [48]; Shakespeare-Finch 2002 [66]; Thompson 2005 [50]; Uchida 2018 [63]; Zavala 2013 [55]; Zavala 2015 [56]; Zavala 2019 [57] |
| **Qualitative Study (n = 13)** | Brady 2019 [43]; Brodie 2012 [29]; Bochantin 2016 [44]; Helfers 2020 [65]; Landers 2020 [32]; Menendez 2006 [45]; Porter 2016 [31]; Regehr 2005 [27] (a); Regehr 2005 [27] (b); Roth 2009 [28]; Waddell 2020 [51]; Watkins 2021 [33]; Wheater 2017 [34] (13) |
| **Mixed Methods Study (n = 3)** | Karaffa 2015 [30]; Hoven 2009 [62]; Richardson 2016 [69] (3) |
| **Data from ER Only (n = 17)** | Anderson 2011 [54]; Brady 2019 [43]; Comer 2014 [60]; Craun 2015 [46]; Erwin 2005 [59]; Gibson 2001 [53]; Haddock 2016 [40]; Halbesleben 2010 [67]; Johnson 2005 [58]; McCoy 2009 [39]; Ryan 2000 [52]; Sanford 2017 [48]; Shakespeare-Finch 2002 [66]; Thompson 2005 [50]; Zavala 2013 [55]; Zavala 2015 [56]; Zavala 2019 [57] |
| **Spouse/Child Data** | **Spouse/partner (n = 22)**: Brodie 2012 [29]; Bochantin 2016 [44]; Brimhall 2016 [41]; Davidson 2006 [36]; Karaffa 2015 [30]; Kishon 2020 [64]; King 2014 [42]; Landers 2020 [32]; McKeon 2020 [38]; Meffert 2014 [36]; Menendez 2006 [45]; Pfefferbaum 2002 [68]; Porter 2016 [31]; Regehr 2005 [27] (a); Regehr 2005 [27] (b); Richardson 2016 [69]; Roberts 2001 [47]; Roberts 2013 [49]; Roth 2009 [28]; Waddell 2020 [51]; Watkins 2021 [33]; Wheater 2017 [34]<br>**Child (n = 6):** Bochantin 2016 [44] (both spouse and child data); Duarte 2006 [61]; Helfers 2020 [65]; Hoven 2009 [62]; Kishon 2020 [64] (both spouse and child data) Uchida 2018 [63] |

*(Continued)*

**Table 1.** (Continued)

| | |
|---|---|
| Study Design | **Cross Sectional (n = 39):** Anderson 2011 [54]; Bochantin, 2016; Brady 2019 [43]; Brimhall 2016 [41]; Brodie 2012 [29]; Comer 2014 [60]; Craun 2015 [46]; Davidson 2006 [36]; Duarte 2006 [61]; Erwin 2005 [59]; Gibson 2001 [53]; Haddock 2016 [40]; Halbesleben 2010 [67]; Helfers 2020 [65]; Hoven 2009 [62]; Johnson 2005 [58]; Karaffa 2015 [30]; Kishon 2020 [64], Landers 2020 [32]; McCoy 2009 [39]; McKeon 2020 [38]; Menendez 2006 [45]; Pfefferbaum 2002 [68]; Porter 2016 [31]; Regehr 2005 [27] (a);Regehr 2005 [27] (b); Richardson 2016 [69]; Roth 2009 [28]; Ryan 2000 [52]; Sanford 2017 [48]; Shakespeare-Finch 2002 [66]; Thompson 2005 [50]; Uchida 2018 [63]; Waddell 2020 [51], Watkins 2021 [33] Wheater 2017 [34] Zavala 2013 [55]; Zavala 2015 [56]; Zavala 2019 [57]<br>**Two time points/longitudinal (n = 4):** Meffert 2014 [36]; King 2014 [42]; Roberts 2001 [47]; Roberts 2013 [49] |
| Topic | **Spousal Mental Health and Wellbeing**<br>Partner distress/mental health (n = 12): Davidson 2006 [36]; Kishon 2020 [64]; Landers 2020 [32]; McKeon 2020 [38]; Meffert 2014 [36]; Menendez 2006 [45]; Pfefferbaum 2002 [68]; Regehr 2005 [27] (a); Regehr 2005 [27] (b); Richardson 2016 [69]; Roberts 2001 [47]; Roberts 2013 [49]; Waddell 2020 [51]; Wheater 2017 [34]<br>Secondary Trauma (n = 2): Landers 2020 [32]; Wheater 2017 [34]<br>**Couple Relationships**<br>Marriage/divorce (n = 2): Haddock 2016 [40]; McCoy 2009 [39]<br>Couple relationships/impact of ER role (n = 15): Brimhall 2016 [41]; Brodie 2012 [29]; Craun 2015 [46]; Halbesleben 2010 [67]; Karaffa 2015 [30]; King 2014 [42]; Meffert 2014 [36]; Porter 2016 [31]; Roberts 2001 [47]; Roberts 2013 [49]; Roth 2009 [28]; Sanford 2017 [48], Thompson 2005 [50]; Waddell 2020 [51]; Watkins 2021 [33]<br>Domestic Violence (n = 9): Anderson 2011 [54]; Erwin 2005 [59]; Gibson 2001 [53]; Johnson 2005 [58]; Meffert 2014 [36]; Ryan 2000 [52]; Zavala 2013 [55]; Zavala 2015 [56]; Zavala 2019 [57]<br>**Child Mental Health and Wellbeing (n = 7)**<br>Brady 2019 [43]; Comer 2014 [60]; Duarte 2006 [61]; Helfers 2020 [65]; Hoven 2009 [62]; Kishon 2020 [64]; Uchida 2018 [63]<br>**Family Coping and Support (n = 10)**<br>Brady 2019 [43]; Brodie 2012 [29]; Karaffa 2015 [30]; Menendez 2006 [45]; Pfefferbaum 2002 [68]; Regehr 2005 [27] (a); Regehr 2005 [27] (b); Richardson 2016 [69]; Roth 2009 [28]; Shakespeare-Finch 2002 [66] |

[a] (Numbers in brackets denote sum of studies in that criterion).

## Conceptual mapping–research themes

**Theme 1 –spousal/partner mental health and wellbeing.** *Pressure experienced by spouses/partners as an ER family.* Nine papers reported that spouses felt extreme pressure due to the nature of the ER day-to-day role and its effects on family life [26–33]. Spouses of US police officers, paramedics and firefighters reported feeling like it was their sole responsibility to manage the household and children, and often described feeling like a single parent [28–30,33]. These descriptions of stressors were reported cross-culturally, spanning different countries and ER branches. Negative issues affecting family life such as long working hours, unpredictable shifts, reduction in quality relationship time, lone parenting and inequitable distribution of family responsibilities were reported in South African spouses with partners in the emergency services [34] and spouses of Canadian paramedics and firefighters [26,27]. Five studies consistently found that spouses of ERs in the US and Canada worried about the danger involved in their partners' ER occupations [26–28,31,32].

Three studies reported specifically on the mental health and wellbeing of spouses/partners of ERs. One Australian police study found 14.0% of spouses reported probable PTSD and

reported distress related to social dysfunction and somatic symptoms [35]. In Meffert, Henn-Haase [36] study, US police spouses reported only mild symptoms of STS (average score 26.8 using Modified Secondary Trauma Questionnaire [37]), however Landers, Dimitropoulos [32] found qualitative evidence of STS among the spouses of US police officers, with spouses describing nausea, intrusive thoughts, and fear and anxiety in relation to the occupational trauma experienced by their police partners.

One Australian study investigated the health of informal caregivers of ERs, which included spouses/partners, family members and friends. They found that the informal caregivers reported higher levels of psychological distress, depression, anxiety, poorer quality of life, worse sleep and lower levels of physical activity compared to the general population norms [38]; however, this study may be limited by a small sample size of n = 30.

**Theme 2—couple relationships.** *Divorce in ER populations.* A US study of police families from 2000 Census data found that divorce/separation rates were lower than the US national average (14.47% v 16.96%), as well as lower than the rate expected given the demographic and income characteristics of the law enforcement workers (14.47% v 16.35%) [39]. A US study of firefighters that asked about previous divorce, found that male firefighters had a higher age standardised prevalence of being currently married, compared to general public census data (77.0% v 58.5%), whereas the divorce rate prevalence was slightly higher (11.8% v 9.4%). For female firefighters, a different pattern was seen. Female firefighters had a lower prevalence of current marriage (42.6% v 55.4%) and had a higher prevalence of divorce compared with comparative female census data (32.1% v 10.0%) [40]. Overall, there was little evidence to compare divorce outcomes.

*Relationship communication and withdrawal.* Thirteen studies found a relationship between quality of communication and ERs spouses/partners relationship experience and concurrent impacts on spouses/partners wellbeing. In a US based study, Brimhall, Bonner [41] found that 24.0% of partners and 29.0% of police officers reported psychological distress due to their relationship. In this study those who perceived secure attachment with their police partner experienced increased mutually constructive communication. Secure attachment perceived by both parties was also associated with increased marital satisfaction [41].

Evidence of emotional withdrawal by either party in the relationship was found to negatively affect spouses/partners' wellbeing and increase marital tension [33,35,42,43]. There are different factors found in these studies that account for why relationship withdrawal might happen in ER relationships, including: withdrawal behaviour associated with PTSD/STS; partners protecting each other from difficult experiences to reduce conflict; behaviours learnt by ERs in their roles causing withdrawal in relationships (e.g., being unemotional and detached from work); and the impact of work stress or shift work and ER/spousal withdrawal.

In a qualitative Canadian study, partners of paramedics reported changes in the personality of their paramedic partner who became closed off and withdrawn as a result of trauma experienced [26]. Bochantin [44] found that US firefighters withdrew from their spouses/partners emotionally as a way of trying to protect their partner or their family. This finding was replicated in a study of 9/11 firefighters whose partners reported their withdrawal and lack of sharing of their work at Ground Zero [45]. Spouses reported frustrations in one study where their police partners only shared limited information about their daily work [29]. A study of US firefighters and their spouses found a negative impact of ER shift work/sleep loss on relationship communication and emotional availability of the ER partner [33]. In an Australian study of police officers, Davidson, Berah [35] found that withdrawal behaviours associated with PTSD was associated with psychological distress in their partners. Meffert, Henn-Haase [36] found US police spousal perceptions of their partners PTSD symptoms was important where those spouses who experienced higher levels of STS and distress had more difficulty in attributing

their partners withdrawal/avoidance behaviours to their PTSD symptoms. Some studies identified that this withdrawal was as a result of behaviours learnt in police job roles, such as being detached and unemotional. Two studies found police child abuse officers reported being more withdrawn in their relationships as a result of their work [43,46]. Karaffa, Openshaw [30] found that US police spouses believed that their partners' job caused them to become unemotional and detached. Work stress was often implicated as a factor that hindered relationship communication and promoted withdrawal. Roberts, Leonard [47] found that increased job stress decreased police officer's awareness of their spouses' negative feelings, whilst spouses concurrently became more attuned to their police partners' hostility, creating poorly functioning communication patterns. King and DeLongis [42] found that work stress increased symptoms of burnout and rumination amongst Canadian paramedics which then caused spouses to withdraw, creating heightened levels of marital tension. Sanford, Kruse [48] also found that US firefighters' job stress was negatively correlated with relationship satisfaction and positively correlated with negative couple behaviours such as withdrawal, blaming or hostility.

*Work stress spillover and compensatory emotional regulation.* Eight studies found that partners of ERs subdued their own emotional needs, avoided conflict and compensated emotionally in their relationships to balance the high stress or mental health problems experienced by their ER partner. For example, Davidson, Berah [35] found that Australian police who scored highly for hyperarousal had partners who scored low on arousal. Roberts and Levenson [49] found that, on days that police experienced high stress, their partners regulated their own emotions to avoid conflict. This study also reported that police job stress created an environment for future marital distress and toxic marital interaction. Landers, Dimitropoulos [32] identified that police spouses reported taking on more responsibilities to ease the burden on their police partners exposed to trauma. In a study of Australian policewomen, Thompson, Kirk [50] found that work stress spilt over into family life through burnout/emotional exhaustion, which decreased family cohesion, whilst at the same time decreasing family conflict because of emotional withdrawal patterns. Police, fire service and ambulance spouses were found to protect their ER partners by not sharing their fears [30,44] and spouses reported feeling the weight of needing to relieve stress in the family [31,51]. Overall studies found that, whilst compensatory emotional regulation by the spouse/partner may reduce marital conflict, it placed a huge emotional burden upon them causing withdrawal and negative wellbeing outcomes.

*Domestic Violence/ Intimate Partner Violence.* Ten papers reported on Domestic Violence (DV)/Intimate Partner Violence (IPV). All studies reporting on DV/IPV were US police studies, and all took data from ERs themselves or from case reports (rather than from ER families directly). Estimates of self-reported physical aggression against a spouse/partner (slapping, punching, injuring, losing control and becoming physically aggressive) ranged from 7.4% - 10.0% [52–57], and 8.1–8.9% for physical aggression against children [53,55].

Three papers reported that police officers who exhibited 'authoritarian spillover' (using authority and command from their role as police officer in domestic settings) or the 'desire to be in control' were more likely to perpetrate IPV than those who did not display authoritarian spillover [54,57,58]. Two papers suggested that job stress [54] or exposure to violence [58] increased authoritarian spillover which increased the likelihood of ERs perpetrating IPV. Furthermore, Erwin, Gershon [59] found increased reports of IPV perpetration in officers' records who worked in high crime precincts, possibly lending support to the relationship between exposure to violence and IPV perpetration. Other studies have suggested that pre-policing events affect police officers' likelihood of IPV perpetration. For example, one study reported that police officers who experienced child maltreatment were more likely to perpetrate IPV [56]. Zavala [55] found that police officers physically abused in childhood, compared to those who were not, were three times more likely to engage in IPV against their spouse/partner and

four times more likely to be violent towards their child. Gibson, Swatt [53] also found an association of physical abuse in childhood and IPV in adult relationships. Meffert, Henn-Haase [36] found that spousal perception of PTSD symptoms in their police partner was associated with total couple violence and spousal/partner to officer violence (i.e., officer victimisation), in that increased reporting of a partner's PTSD symptoms by the spouse was associated with increased violence in the relationship.

**Theme 3—child mental health and wellbeing.**  Seven studies assessed the mental health and wellbeing of children from ER families. Six studies came from the US and one study from Israel.

Comer, Kerns [60] found 10.8% children who had an ER parent or relative involved in the Boston Marathon bombing manhunt reported probable PTSD compared to 1.9% of children who did not have an ER parent or relative. Two studies found that children of EMTs who had a World Trade Center (WTC) role during the terrorist attacks reported the highest probable PTSD (18.9% and 15%), followed by children with a police parent (10.6% and 8%) and then by those with a firefighter parent (5.6% and 3%) [61,62]. Some of these differences were accounted for by demographics such as race and ethnicity, with poorer non-white families being over-represented in US EMT populations, but Duarte, Hoven [61] also found if there were two ERs as parents or in the wider family, then the prevalence of PTSD increased to 17.0%. Lastly Duarte, Hoven [61] found that levels of direct child exposure to the WTC terrorist attack to be a confounding variable which increased probable PTSD rates to 35.6% in children who had two ER parents.

A third WTC US study confirmed that trauma symptomology in police impacted their children's mental health and was associated with behavioural difficulties. Uchida, Feng [63] found that police parent dysphoric arousal symptoms were associated with their children being more fearful and clingy, presenting with more externalising behaviours and increased somatic problems. The prevalence of child behavioural problems was reported at 20.0% for police responder children and 31.4% for non-traditional responder children populations (such as construction workers involved in the WTC site). Uchida, Feng [63] argued that, whilst PTSD in police parents has a large impact on their children's health, it may also be that ER families are more accustomed to coping with trauma and therefore experience fewer negative outcomes compared to other families.

In a study of Israeli children of ERs and control group of non ER families, Kishon, Geronazzo-Alman [64] found overall that paternal exposure to traumatic events was significantly related to child PTSD symptoms; however, in final models there was a non-significant relationship of paternal ER status and child PTSD symptoms. The authors posit this finding may be due to more firefighters in their sample whose children may have lower levels of PTSD compared to other ER branches, supporting previous studies [61,62]. The study also identified that maternal occupational exposure was significantly associated with a higher number of symptoms of anxiety, depression and other mental health disorders in children even though participating mothers were not ERs.

Four qualitative studies examined why children of ERs may have adverse mental health and wellbeing outcomes. Brady, Fansher [43] found US police working in child abuse investigation roles reported strained and overprotective relationships with their children, as well as reporting less time and empathy with their children because of their police roles. Helfers, Reynolds [65] investigated the lived experiences of children of police officers. They found children reported overprotective parenting, harassment and bullying from others because of their parent's occupation, and worry for the safety of their police parent negatively affecting their wellbeing. Menendez, Molloy [45] described anxiety in children of firefighters involved in the WTC event, in relation to increased concerns for the safety of their ER parent and attending

funerals for ER families involved in the WTC event. In a more general ER context, Bochantin [44] described the burden of emotional labour taken on by the children of police and fire personnel who often concealed their emotions by putting on a 'brave' face, whilst simultaneously being concerned for the safety of their ER parent.

**Theme 4—family support from the organisation and coping strategies.** Little or no government and organisational support for families was reported by Australian ER spouses/ partners, US police spouses/partners, and Canadian paramedic and firefighter spouses/partners [26,27,30,51]. Australian ER spouses/partners in one study living with ER partners with PTSD reported a sense of loneliness and difficulty in finding peer support with people who understood PTSD [51]. Five studies highlighted how crucial informal social support was for ER spouses/partners and families to deal with day-to-day pressures of ER life [28–30,43,66]. One study found in police couples that where spousal support was high, individuals were more likely to use active coping (problem focused methods to regain control over stress) compared to avoidant coping (avoiding sources of stress), and this positive relationship was stronger for couples working in the same workplace, such as dual serving police couples [67]. Shakespeare-Finch, Smith [66] found that Australian ambulance personnel used more varied coping strategies compared to the general population, including self-care, social support and rational cognitive coping that all had positive impacts on ER family functioning.

Three studies emphasised the positive impact of social support in the aftermath of bereavement in ER families [45,68,69]. Richardson [69] found over the course of ten years that social support amongst 9/11 firefighter widows played a lasting and enduring role in their recoveries. Fifty-eight percent of the sample endorsed that informal support had helped with their healing and 60.0% of the sample still met with other widows ten years later. Richardson [69] also investigated Post Traumatic Growth (PTG) in this sample. PTG has been defined as the experience of positive change after a traumatic experience, including deeper connection with others, finding new meaning in life and spiritual growth [70]. Richardson [69] found PTG was correlated with ERs widows use of New York Fire department sponsored support groups, one-on-one therapy, socialising with other widows and local support groups. Menendez, Molloy [45] found that 9/11 firefighter widows utilised social support from other widows to help them deal with their anxiety, fears and sadness. Other studies have highlighted the positive effect of spiritual beliefs in helping ER families cope with trauma [30,45] and the use of humour to aid day-to-day coping [29,44].

**Theme 5—positive experiences of ER families.** Whilst investigation of positive aspects of being part of an ER family is limited, eight US studies and one South African study examined this topic in some way. Some of these papers highlighted the pride that spouses/partners and children of ERs felt in being part of a responder family and the positive impact that their family member had on the community [30,31,34,43]. Children of police officers described the benefits of their police parent in terms of protective parenting that made them feel safe and the positive resource their parent was in giving them knowledge about their rights and legal protections [65]. Police spouses in Brodie and Eppler [29] described the benefits of the police career choice such as camaraderie, helping others, pride felt in the job, financial security and health insurance benefits. Some studies also reported that, despite the huge pressures faced by ER families, they often presented as highly resilient and functioning [28,63,69]. This finding aligns with Shakespeare-Finch, Smith [66] findings that ambulance personnel deployed more varied coping strategies with their families which improved resilience and family functioning through this experiential learning.

## Discussion

Our systematic review identified 43 papers that addressed the mental health and wellbeing of ER families. We frame the proceeding discussion around our four review objectives which were to investigate types of mental health and wellbeing outcomes investigated, prevalence of positive and negative mental health outcomes, experiences, and risk and protective factors for ER family's mental health and wellbeing.

**Types of mental health and wellbeing outcomes.**   The main types of mental health and wellbeing outcomes investigated in family members of those in ER roles included: 1) Spousal/partner mental health and wellbeing–specifically PTSD, secondary traumatic stress and wellbeing issues associated with ER occupational burdens on family life and concern for the safety of their ER partner; 2) Couple relationships–specifically evidence was found on relationship communication, work-stress spillover and limited evidence found concerning divorce and DV/IPV; 3) Child mental health and wellbeing–evidence was found on the impact of primary and secondary trauma from ER parent roles on children and concerns children had for the safety of their ER parent; 4) Family support and coping strategies–findings mainly focused on benefits of social support for ER families; and 5) Positive experiences of ER families–findings found positive attributes of the ER occupation that improved ER families wellbeing and esteem, for instance, pride in the ER role and camaraderie.

**Prevalence of mental health and wellbeing outcomes.**   In terms of assessing the prevalence of mental health and wellbeing outcomes in ER families, only nine papers reported specifically on spouses/partners/children's mental health and wellbeing levels in specific cross-sectional samples, offering little evidence on which to assess prevalence outcomes.

Only one study measured probable PTSD in police spouses [35], one study measured STS [36] and two studies measured psychological distress in police partners and ER informal caregivers [38,41]. It is therefore difficult to assess ER spouses/partners mental health or wellbeing outcomes compared to general population levels. Military families may be a useful basis of comparison to ER families in that they experience a partner who has exposure to trauma and a high tempo job, where occupational and emotional demands may impact on the family [71]. Whilst not all aspects of military life may be directly translatable, such as factors like deployments with longer separations, more military families are living dispersed within communities [72] and their experiences may shed light on understanding potential mental health and wellbeing risks in ER families. For example, there is evidence found in military family research, to suggest differential mental health and wellbeing outcomes compared to general populations. For example, US military research finds a third of junior military spouses screened positive for one or more psychiatric conditions [73]. UK military research found that women military spouses/partners had a significantly higher prevalence of probable depression, hazardous alcohol consumption, and binge-drinking compared to women in the general population [8].

Studies conducted with children of ERs found probable PTSD ranging from 2.9–18.9%; however, these studies were focused on mass trauma events such as the WTC attacks and the Boston marathon bombing where it is difficult to delineate the impact of direct/indirect traumatic exposures and the impact of being part of an ER family. In three of these WTC studies, children of EMT's reported higher levels of probable PTSD compared to the general population, but police and fire ER children reported similar or lower levels of probable PTSD compared to the general population [61–63]. This may indicate children's comparative resilience in these ER families compared to other families during these mass trauma events, although it is difficult to make robust conclusions from these cross-sectional studies and gives limited information on the day-to-day occupational exposures impact on ER families.

**Family experiences, risk and protective factors.**  In terms of factors influencing ER family mental health and wellbeing, we found evidence that the nature of the ER occupation impacts the family directly and indirectly through day-to-day occupational stressors and effects of trauma exposure. Protective factors for family mental health and wellbeing included social support, organisational support and adaptive coping strategies.

**Occupational stressors risk factors.**  *ER day-to-day occupation burden on spouse.* The systematic review studies identified an impact on spousal/partner and child health from the day-to-day pressures of the ER occupation. From the available studies there is burgeoning evidence of the ER burden of work-stress spillover that impacts family mental health, wellbeing and family functioning. The evidence describes the stressors and burden of responsibility experienced by spouses/partners in running the household, often feeling like a lone actor/parent because of the demands of the ER occupational role [26–30,33]. Other occupations that report non-standard working times, such as evenings, nights and weekends, akin to many ER shift patterns, also find individuals report time-based work family conflict, where the working partner cannot be around to partake in family activities, leaving the other partner with the family responsibilities [74]. Although the nature and frequency of familial separation and strains of ongoing micro-transitions seem distinct for families of ER, this description of family life and pressures experienced may be similar in some respects to military spouses/partners who describe lone parenting and episodes of separation and disruption to family life due to deployments [71].

*ER Work-Stress Spillover.* The review evidence identified work-stress spillover that impacted couple relationships, decreasing family cohesion [50], where spouses often purposely took on more practical and emotional family tasks to alleviate their ER partners' stress and avoid conflict [31,33]. Research has found that uneven household labour, where spouses felt disproportionately responsible for household management and child adjustment, was associated with strains on mothers' personal wellbeing and lower satisfaction with their relationship [75]. This ER occupational burden on spouses may have negative long-term personal and relational impacts. ER Work-Family Conflict (WFC) literature provides confirmatory evidence of the spillover stress model identified in this review. WFC is the concept that describes and measures the tension between work and family arenas, where work demands interfere with family needs with potential negative outcomes for family life and functioning [76]. ER WFC research has found that the ER occupation interfered with the time and emotional energy the ER had which impacted on spousal distress and negatively affected the way the ERs parented and interacted with their children [77,78]. General population research has identified that perceptions of spouses' work to family conflict were associated with spousal mental health, distress, anger and their children's problems with friends, health and school [79]. The impact of the pressures of the ER occupation on the family may have wider implications for spouse's parental stress and concurrently impact on their children's wellbeing. Parenting stress has been identified to negatively impact parenting behaviours which effects children's development, social competence and internalizing/externalising behaviours at pre-school [80,81].

*Domestic Violence/Intimate Partner Violence.* Work-stress spillover or authoritarian spillover were associated with DV/IPV in police families in this review [54,57,58]. There was no evidence on the experiences of other branches of ER families. This area is understudied with no data taken from spouses/partners or children themselves in the review papers. Some ER roles may share similar characteristics to military occupations where IPV perpetration is prevalent [82], such as a culture of macho environments, verbal aggression and command and control behaviours practiced occupationally [83–85]. If these behaviours cascade over into family life as identified in the review, they may in turn be problematic for aggressive family behaviours [58]. Mazza, Marano [86] highlight that in situations where danger is present, anxiety

and fears may lead to feelings of impotence, and in response to threats, aggression and IPV can be perpetrated, hence it is possible ER families may experience this cycle with their ER partner. It is unknown from the available literature whether ER professions with more of a gender balance and caring vocations such as paramedics may have less aggressive spillover behaviours into the family environment. Therefore in lieu of this evidence, DV/IPV in these professions needs to be a future area of investigation.

**Trauma exposure risk factors.** *Family concern and distress for ER safety*. The qualitative literature in the systematic review begins to build an evidence base that highlights direct and indirect trauma risk factors that negatively impact on spouses/partners' wellbeing. One of these factors is the concern and anxiety spouses have for their ER partner's safety because of dangerous occupational exposures. This again is similar to concerns in research literature highlighted by military spouses for the welfare of their spouses/partners whilst deployed and negative effects on spousal wellbeing [6]. The review found that these worries were also held by children of ERs who had general anxiety about their ER parent's job role and dangers experienced [44,65]. Additionally Helfers, Reynolds [65] found children of police officers reported harassment and bullying by others who did not support the policing community. Both worry and harassment/bullying in children have been shown to negatively impact stress, mental health outcomes, function and quality of life, and hence should be areas of investigation for ER children [87–89].

*ER partner with mental health problem impact on spouse/partner*. Two studies in the review identified that ERs with PTSD may negatively impact the mental health and wellbeing of their spouse/partner or informal carer [35,38]. As ERs have been evidenced to experience higher levels of mental health problems than the general population [1–3], spouses/partners of ERs are potentially exposed to more risk. Whilst there is little ER quantitative research in this area, this finding does align with a previous systematic review that found a systemic negative impact of trauma survivors' PTSD on the psychological distress of their intimate partners [90]. More recent research has found that partners of military personnel with PTSD also have poor mental health outcomes themselves [91,92]. A systematic review found there was evidence of secondary traumatic stress in partners of help-seeking veterans with PTSD [93], and a scoping review identified negative impacts on military spouses/partners associated with a military partner's operational stress injury, such as PTSD, depression or any psychological difficulty due to service [94]. An additional military study has found these stressors may be particularly impactful at times of transition from service [95], which may be similar for ER's leaving their professions.

*Poor Relationship Communication*. The papers in this review assessing couples' relational experiences found evidence of poor communication and withdrawal in ER couples due to ERs' PTSD or protective behaviours to shield each other from trauma. There is much evidence that good communication within relationships is protective of relationship satisfaction and secure attachment [96] and confirmatory evidence that disclosure of traumatic experiences can reduce distress within relationships [97]. The behaviours seen in ERs and their partners which aimed to reduce communication to protect each other, may be self-defeating. Additionally, research has found that positive partner communication mediates the association between trauma symptoms and relationship satisfaction [98]. There may however be extra barriers in ER occupations of communication where they are not at liberty to share details of experiences due to confidentiality. Hence, supporting ER couples to engage in effective communication, where issues and emotions are discussed without breaking confidentiality, could be protective and may be particularly pertinent if the ER partner experiences mental health problems. A review of couple relationship education studies showed that the large majority of studies where couples participated in relationship education compared to controls improved their

communication, relationship satisfaction and stability [99]. ER organisations or voluntary sector support organisations could consider relationship training education best practice guidelines suggested in Bakhurst, Loew [100] and adapt these to ER families experiences. The review, however, found a lack of evidence regarding the prevalence of relationship satisfaction in ER couples which is an important variable to assess long-term relationship trajectories and their relationship with work-stress spillover [101,102]. Finally, the review papers did not assess the sexual dimension of couple relationships and how intimacy and relationships are affected by work burdens, time spent together and distress, hence this is an important factor that should be taken into account with regards to ER relationships and wellbeing [103].

*ER Parent with Mental Health Problem impact on child.* There is some evidence in the review that children's mental health and wellbeing may be affected by their ER parents' PTSD (or their parent's involvement in traumatic events) [63,64]. General population studies have found maternal PTSD to negatively affect child development through parenting stress [104,105]. These findings support one study in this review that found maternal occupational exposure to trauma, in the partner who was not in the ER occupation, impacted child mental health more than ER paternal exposure [64]. This identifies the importance of spousal/partner mental health in impacting children and the need to assess families as a unit. Other populations have evidenced the association of paternal PTSD and adverse child outcomes, including research in Vietnamese child refugees that found a higher risk of mental ill-health associated with paternal PTSD [106], and research from military families which found impacts of paternal PTSD on negative childhood emotional and behavioural well-being outcomes [107]. A scoping review identified multiple impacts on the family and children of Canadian military personnel who experienced persistent psychological difficulties as a result of operational duties in the Armed Forces [108]. Our systematic review found less data with regards to children's general day-to-day experience of their parent's ER occupation and parenting. Understanding this experience is important as a review of military-connected children found their mental health may be affected by family separation, mobility, and demands of the military occupation with regards to physical and mental health [7]. It is therefore pertinent that data is obtained from children themselves where possible and from spouses/ER parents regarding their children, as well as other observers of behaviours like school teachers.

**ER family support protective factors.** The review found consistent evidence for the positive impact of social support as a coping strategy to mitigate the impact of the effect of the ER occupation on the family from stress, trauma and bereavement. This finding is replicated in many other research areas. Sippel, Pietrzak [109] provide evidence for the critical nature of social support and social networks in bolstering individual and family resilience in trauma-exposed groups. Michalopoulos and Aparicio [110] identified that social workers reported reduced vicarious trauma when they had increased social support. Tam-Seto, Krupa [111] highlighted the importance of social support for military families but also found that families in some environments felt stigmatised for seeking support. ER WFC literature highlights the importance of organisational leadership and support in promoting family-friendly policies which in turn positively impact family resilience and wellbeing [112]. As this review found ER families reported little to no organisational support for their wellbeing needs, this is a practical area that ER organisations might focus efforts on to improve family wellbeing and create additional positive organisational impacts on ER job satisfaction and retention [50,113–115].

Finally, a minority of review papers reported evidence of positive outcomes or attributes of ER family life such as family pride in the ER job role and benefits of camaraderie. However, these studies did not directly assess how these positive aspects of being part of an ER family impacted on family wellbeing. Understanding and measuring positive aspects of ER occupation on the family should be a focus for research in the future to understand what factors

protect families from adverse outcomes and what factors improve ER family resilience and wellbeing.

**Limitations of papers.**   The systematic review has highlighted limitations of this body of research literature. When looking at studies characteristics (Table 1), studies have primarily been conducted in the US which lessens the extent to which we can generalise about ER family experiences internationally. Whilst it is not unexpected that the majority of studies focus on police families, as often they are the largest branch of ERs in many countries, there is a concurrent lack of research assessing paramedics/emergency medicine technician families which should be a target for research in the future.

Methodologically the majority of studies were cross-sectional in nature which limits understanding of causation and the specific impact of being within an ER family. Longitudinal research should be conducted to understand broader mental health outcomes in ER spouses/partners/informal caregivers and children, assessing ER occupational stressors as well as specific events that could impact ER families. Studies were also primarily quantitative in nature; however, there was a lack of research assessing ER spouse/partner and children's mental health outcomes as measured quantitatively by validated measures as a primary concern. We know little for example about CMD, PTSD and STS in ER spouses and no studies were identified assessing alcohol use. Research could usefully focus on the mental health and wellbeing of children of ERs including impact of trauma and more generally from their experience of their parent's ER occupation and parenting. Other research gaps were identified such as limited information on relationship satisfaction, sexual intimacy, DV/IPV, ER family experiences of harassment/bullying or misunderstanding from the public, positive experiences of ER families, landscape of support services, and no studies were identified that examined ER families' experiences of transition and post-service. Whilst this review was conducted during the COVID-19 pandemic, we did not find studies addressing the impact of the pandemic on ER families, this specific impact may be important to understand whether there are long-term effects on ER families. It should be noted there were good qualitative studies that provided rich data within this review, but they were limited in number and therefore both quantitative and qualitative studies should be encouraged in the future, focused on current gaps in research.

A large minority of research papers identified in this review focused on data from the ER themselves, typically addressing family issues in terms of their effect on the ER rather than conceptualising the family as an interdependent unit. This approach is problematic when aiming to assess the family health and wellbeing issues of DV/IPV or if only paternal mental health is assessed without recognition of the potential impact of maternal mental health. This limitation is not unusual in family research and has been highlighted in other occupational military research [8]. Data should therefore be collected from spouses/partners/informal carers and children themselves. We also encourage researchers to conceptualise the ER family within a social-ecological model [116], where relationships affecting public health, resilience, mental health and wellbeing are multifaceted and interrelated between individuals, families, organisations, communities, public policy and government.

**Strengths and limitations of the review.**   Strengths of the systematic review include a robust and replicable search protocol. The synthesis strengthens the work by allowing integration of both quantitative and qualitative research thematically. The systematic review has limitations in the extent of extractable quantitative data returned which meant meta-analyses were not possible. The review may also be limited by the databases utilised for searches and in aiming to cover a wide range of mental health and wellbeing search terms, it is possible more specific or nuanced areas of investigation were not captured in depth.

## Conclusions

There is some evidence of the negative impact of the ER occupation on the mental health and wellbeing of ER families; however, this evidence is limited both in volume, topic areas and methodological approaches. More quantitative and qualitative research is need internationally to provide robust evidence on prevalence of outcomes, associated risk/protective factors and experiences of ER families. This area of research is important to improve knowledge of baseline needs to be able to target interventions to support ER families. Longer term, supporting ER family's needs may have a positive impact in improving family functioning and mental health, public health, and increasing job satisfaction, retention and operational effectiveness of ER personnel.

## Supporting information

**S1 Prisma checklist.**
(DOC)

**S1 Table. Full table of study characteristics and findings.**
(DOCX)

**S1 Protocol.**
(PDF)

**S1 File. Search definitions and criteria.**
(DOCX)

**S2 File. Full list of search terms.**
(DOCX)

**S3 File. Quality analysis method.**
(DOCX)

## Author Contributions

**Conceptualization:** Marie-Louise Sharp, Virginia Harrison, Graham Pike, Nicola T. Fear.

**Data curation:** Marie-Louise Sharp, Noa Solomon.

**Formal analysis:** Marie-Louise Sharp, Noa Solomon.

**Funding acquisition:** Marie-Louise Sharp, Nicola T. Fear.

**Investigation:** Marie-Louise Sharp, Noa Solomon, Nicola T. Fear.

**Methodology:** Marie-Louise Sharp, Virginia Harrison, Heidi Cramm, Graham Pike, Nicola T. Fear.

**Project administration:** Marie-Louise Sharp, Nicola T. Fear.

**Resources:** Nicola T. Fear.

**Supervision:** Marie-Louise Sharp, Nicola T. Fear.

**Validation:** Marie-Louise Sharp.

**Writing – original draft:** Marie-Louise Sharp, Noa Solomon.

**Writing – review & editing:** Marie-Louise Sharp, Noa Solomon, Virginia Harrison, Rachael Gribble, Heidi Cramm, Graham Pike, Nicola T. Fear.

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
