## [Decision Letter · Decision Letter 0]

22 Feb 2022

PONE-D-21-23204The mental health and wellbeing of spouses, partners and children of emergency responders: a systematic reviewPLOS ONE

Dear Dr. Sharp,

Thank you for submitting your manuscript to PLOS ONE. After careful consideration, we feel that it has merit but does not fully meet PLOS ONE’s publication criteria as it currently stands. Therefore, we invite you to submit a revised version of the manuscript that addresses the points raised during the review process.

We look forward to receiving your revised manuscript.

Kind regards,

Marianna Mazza

Academic Editor

PLOS ONE

https://journals.plos.org/plosone/s/fileid=ba62/PLOSOne_formatting_sample_title_authors_affiliations.pdf".

“I have read the journal's policy and the authors of this manuscript have the following competing interests: NF is part funded by a UK Ministry of Defence grant, is a trustee of The Warrior Programme - a charity supporting the wellbeing of veterans, service personnel and their families, a specialist academic member of the Independent Group Advising UK NHS Digital on the Release of Patient Data and Chair of the Royal Foundation Emergency Responder Senior Leader Board”

Reviewers' comments:

Reviewer's Responses to Questions

**Comments to the Author**

1. Is the manuscript technically sound, and do the data support the conclusions?

Reviewer #1: Yes

Reviewer #2: Yes

2. Has the statistical analysis been performed appropriately and rigorously? 

Reviewer #1: Yes

Reviewer #2: Yes

3. Have the authors made all data underlying the findings in their manuscript fully available?

Reviewer #1: Yes

Reviewer #2: Yes

4. Is the manuscript presented in an intelligible fashion and written in standard English?

Reviewer #1: Yes

Reviewer #2: Yes

5. Review Comments to the Author

Reviewer #1: The authors are addressing an important and timely topic in the field of mental health. I would like to see the discussion more focused on interpreting study results.

It is important recognizing the significance of the sexual dimension in couple relationship. Other key references and reviews also need to be included. It may be useful to read the following papers:

Marano G, Traversi G, Mazza M. Web-mediated Counseling Relationship in Support of the New Sexuality and Affectivity During the COVID-19 Epidemic: A Continuum Between Desire and Fear. Arch Sex Behav. 2021 Apr;50(3):753-755. doi: 10.1007/s10508-020-01908-3.

Mazza M, Marano G, Lai C, Janiri L, Sani G. Danger in danger: Interpersonal violence during COVID-19 quarantine. Psychiatry Res. 2020 Jul;289:113046. doi: 10.1016/j.psychres.2020.113046.

Once again, I appreciate your efforts in doing this study and writing up this article and wish you the very best.

Reviewer #2: Thank you for submitting this article. I thought it was an engaging article, highlighting mental health and well-being in ER families. This s/review identifies many research gaps, where industry can better support their employees and esp. that vicarious trauma requires much more research. Most of my feedback is minor in nature; the three main comments pertain to: In-text referencing which does not appear correct; some comparisons to military families; size of appendix 1. Comments can be found below:

Feedback PONE D-21-23204

Please check-text referencing throughout.

Vancouver in-text referencing style:

Jones and Jones (ref ppx-x) states that…

Jones, Jones and Jones research determined…

Jones et al (ref) found…

Line 48

Written as ‘compared the general population’

Should be ‘compared to the general population’

Lines 75-76

Research question (2) clarification

The prevalence of mental health and wellbeing is 100% as everyone has ‘mental health’ and ‘wellbeing’. Are you looking for positive MH and WB or adverse? These needs clarification in the research question.

Lines 77-78

Research question (3) clarification

Suggest ‘what are the experiences of ER spouses/partners and children’ and ‘risk and protective factors’ be 2 separate research questions as they are 2 separate concepts. Alternatively, make risk/protective factors (3a).

Line 105 / Appendix 1

Appendix 1 – did you only include full text articles?

Line 123 + 144

Please clarify external stakeholders- who are they and why was there a need to have them review articles for inclusion? This needs elaboration.

Line 168 / Figure 1

Suggest the box ‘131 articles excluded…’ is moved to horizontal position (like ‘irrelevant records’) rather than portrait- as these articles have been excluded.

Lines 176 -177

The sample from the US should be (n=33) rather than (33) which looks like a reference. Same with other countries.

Table 1

Table 1 essentially repeats the paragraph ‘Overview of studies’ – suggest it can be removed. Alternatively, if a list of identified/included articles is required, a simple table as shown below might suffice.

Citation Country ER type

USAUKAust PFFEMT/P

Jones…* **

Smith…* *

White…* *

(Sorry, the table didn't copy/paste well, but gives the general idea)

Line 205

Secondary traumatic stress should be STS

Line 225 and Line 412

Re ‘little evidence to compare divorce outcomes’ - on line 225 indicates data was equivocal, whereas the wording in line 412 ‘specifically evidence was found on divorce’ suggests unequivocal. Suggest rewording for clarification.

Line 319

WTC abbreviation. I think this is the first time this abbreviation has been used. Please type in full. Also, need an abbreviation index/key somewhere.

Line 390 - Positive experiences of ER families

Compartmentalising / experiential learning effect from ref 65 Shakespeare could be mentioned here or discussed elsewhere. This is a protective mechanism.

Line 405

Current wording - ‘prevalence of mental health…’

See research question feedback

Lines 425-432

ER family and comparison to military family: I can see some similarities, however, if military personnel are deployed overseas (war zone or peacekeeping) there is far less communication between partners/families, and they can be separated for months/years at a time. Military families often live on a military base; therefore, support/environment differs from ER who live scattered in suburbs. This paragraph requires some elaboration to make this comparison appropriate.

Lines 456 – 459.

Another comparison between ER family and military family. I feel military ‘lone parenting’ is the extreme version, as the parent at home is truly alone for however long their partner is deployed. ER families rarely experience this level of lone parenting outside of extreme disasters. Other occupations by which to compare may be more appropriate (Dr? Nurse?)

Lines 486-488

I’m ok with this comparison between ER and military. Perhaps discuss the shift towards more balanced genders in these roles (at least for paramedicine this is true, unsure re other occupations) and whether this might/not change the macho environment etc.

Lines 494 - 496.

I’m ok with this comparison between ER and military. Vicarious trauma is definitely something partners/children, (even researchers and journalists) can experience.

Lines 528-530.

Good. Perhaps discuss the difficulty of confidentiality in these professions. ER (and military) are not at liberty to discuss work activities- this differs from many occupations where you can informally debrief with your loved one after work.

Line 543

I think the word ‘non’ is not required?

Lines 553 – 556.

I’m ok with this comparison to military too. Highlights research gaps.

Line 601

Suggests the research was conducted after the COVID-19 pandemic. I’d argue we are still within the pandemic? Agreed the impact of the pandemic on ER families must be researched.

Line 618-622

Suggest there are other limitations to this systematic review- data bases searched etc, understand word limitations may have restricted explanation.

Appendix 3

Love the detail in this table, however it looks more like the Data Extraction Tool than a succinct summary for a reader. If full data is required to be available, maybe suggest this is supplementary material, and a shorter version supplied for the article?

In addition, there is some inconsistency with reporting data (if staying with more in-depth table). Consider Stanford (ref 47) and compare to Davidson (ref 34). Davidson has quite an in-depth explanation of measures, while Stanford does not.

Also, significant statistics being in bold for Davidson (ref 34) compared to Roberts (ref 48) which is not.

There is a need for an abbreviation index/key in this appendix.

Gibson (ref 52): The strain was largest in BOTH anger and depression?

Haddock (ref 39): Ever divorce? Definition required.

Pfefferbaum (ref 67) Groups A/B/C/D explained in the column after being mentioned previously. Define at first use.

There are a few lines which need proof reading, the one I highlighted: Bochantin (ref 43) ‘This study existing work–family research by moving beyond it being…’.

Richardson (ref 68) Coding revealed 11 themes. Given the extent of your other explanations for other articles, what are those 11 themes?

6. PLOS authors have the option to publish the peer review history of their article (what does this mean?). If published, this will include your full peer review and any attached files.

Reviewer #1: No

Reviewer #2: No

---

## [Author Response · Author response to Decision Letter 0]

21 May 2022

PONE-D-21-23204

The mental health and wellbeing of spouses, partners and children of emergency responders: a systematic review

PLOS ONE

Response to Reviewers

We would like to thank the reviewers for their time and expertise in assessing this work, we have incorporated the changes detailed below (our responses in bold next to your original comments) and believe the manuscript is strengthened by the additions. Kind Regards.

Reviewer #1: The authors are addressing an important and timely topic in the field of mental health. I would like to see the discussion more focused on interpreting study results.

It is important recognizing the significance of the sexual dimension in couple relationship. Other key references and reviews also need to be included. It may be useful to read the following papers:

Marano G, Traversi G, Mazza M. Web-mediated Counseling Relationship in Support of the New Sexuality and Affectivity During the COVID-19 Epidemic: A Continuum Between Desire and Fear. Arch Sex Behav. 2021 Apr;50(3):753-755. doi: 10.1007/s10508-020-01908-3.

Mazza M, Marano G, Lai C, Janiri L, Sani G. Danger in danger: Interpersonal violence during COVID-19 quarantine. Psychiatry Res. 2020 Jul;289:113046. doi: 10.1016/j.psychres.2020.113046.

Once again, I appreciate your efforts in doing this study and writing up this article and wish you the very best.

- Thank you for your feedback. We have expanded points in our discussion to reflect the sexual dimension in couple relationships and to reflect the papers cited above. These read:

Line 490

‘Mazza, Marano (83) highlight that in situations where danger is present, anxiety and fears may lead to feelings of impotence, and in response to threats, aggression and IPV can be perpetrated, hence it is possible ER families may experience this cycle with their ER partner.’

Line 540

‘Finally, the review papers did not assess the sexual dimension of couple relationships and how intimacy and relationships are affected by work burdens, time spent together and distress, hence this is an important factor that should be taken into account with regards to ER relationships and wellbeing (100).’

Reviewer #2: Thank you for submitting this article. I thought it was an engaging article, highlighting mental health and well-being in ER families. This s/review identifies many research gaps, where industry can better support their employees and esp. that vicarious trauma requires much more research. Most of my feedback is minor in nature; the three main comments pertain to: In-text referencing which does not appear correct; some comparisons to military families; size of appendix 1. Comments can be found below:

Feedback PONE D-21-23204

Please check-text referencing throughout.

Vancouver in-text referencing style:

Jones and Jones (ref ppx-x) states that…

Jones, Jones and Jones research determined…

Jones et al (ref) found…

- We have done a proof of the text and checked Vancouver style as created through our referencing software. 

Line 48

Written as ‘compared the general population’

Should be ‘compared to the general population’

- Amended

Lines 75-76

Research question (2) clarification

The prevalence of mental health and wellbeing is 100% as everyone has ‘mental health’ and ‘wellbeing’. Are you looking for positive MH and WB or adverse? These needs clarification in the research question.

Thank you – these questions have been amended to read:

2) What is the prevalence of positive and negative mental health and wellbeing outcomes in ER spouses/partners and children?

Lines 77-78

Research question (3) clarification

Suggest ‘what are the experiences of ER spouses/partners and children’ and ‘risk and protective factors’ be 2 separate research questions as they are 2 separate concepts. Alternatively, make risk/protective factors (3a).

Amended to read:

3) What are the experiences of ER spouses/partners and children with regards to mental health and wellbeing?

4) What are the risk and protective factors that are associated with an ER family’s mental health and wellbeing?

Line 105 / Appendix 1

Appendix 1 – did you only include full text articles?

Yes – we have amended this to reflect that eligibility was based on where full text articles could be accessed.

Line 123 + 144

Please clarify external stakeholders- who are they and why was there a need to have them review articles for inclusion? This needs elaboration.

This has been clarified in lines 127, 129 and 150. External stakeholders were those working in emergency responder organisations or charities. It is part of our patient and public inclusion work to engage those with lived experience to ensure we had included relevant research and reflected the lived experience/operational experience in the interpretation of results.

Line 168 / Figure 1

Suggest the box ‘131 articles excluded…’ is moved to horizontal position (like ‘irrelevant records’) rather than portrait- as these articles have been excluded.

- Amended to a horizontal position

Lines 176 -177

The sample from the US should be (n=33) rather than (33) which looks like a reference. Same with other countries.

- Amended

Table 1

Table 1 essentially repeats the paragraph ‘Overview of studies’ – suggest it can be removed. Alternatively, if a list of identified/included articles is required, a simple table as shown below might suffice.

Citation Country ER type

USAUKAust PFFEMT/P

Jones…* **

Smith…* *

White…* *

(Sorry, the table didn't copy/paste well, but gives the general idea)

- We have left Table 1 as initially suggested. We believe in terms of understanding characteristics of papers, this provides a visualised way to group the studies. We believe listing study by study is not as efficient for readers when trying to assess similar characteristics between studies. If readers want more detail study by study we have provided Appendix Three for in-depth reference.

Line 205

Secondary traumatic stress should be STS

- Amended

Line 225 and Line 412

Re ‘little evidence to compare divorce outcomes’ - on line 225 indicates data was equivocal, whereas the wording in line 412 ‘specifically evidence was found on divorce’ suggests unequivocal. Suggest rewording for clarification.

- This has been clarified to note limited evidence was found on divorce

Line 319

WTC abbreviation. I think this is the first time this abbreviation has been used. Please type in full. Also, need an abbreviation index/key somewhere.

- This has been amended. 

Line 390 - Positive experiences of ER families

Compartmentalising / experiential learning effect from ref 65 Shakespeare could be mentioned here or discussed elsewhere. This is a protective mechanism.

- We have added this extra discussion point to read:

Line 406

‘Some studies also reported that, despite the huge pressures faced by ER families, they often presented as highly resilient and functioning (27, 62, 68). This aligns with Shakespeare-Finch, Smith (65) findings that ambulance personnel deployed more varied coping strategies with their families which improved resilience and family functioning through this experiential learning.’ 

Line 405

Current wording - ‘prevalence of mental health…’

See research question feedback

- amended to reflect ‘positive and negative mental health outcomes’.

Lines 425-432

ER family and comparison to military family: I can see some similarities, however, if military personnel are deployed overseas (war zone or peacekeeping) there is far less communication between partners/families, and they can be separated for months/years at a time. Military families often live on a military base; therefore, support/environment differs from ER who live scattered in suburbs. This paragraph requires some elaboration to make this comparison appropriate.

- We have added a paragraph describing the basis for comparison further and cited articles that directly use this comparison. The differences asserted above have reduced in recent years due to developments in technology creating more effective communications for families, and that more military families are living dispersed in general population communities. In comparison to comparing to general population families, we believe the similar experience of occupational burden and experience to trauma can highlight where risk might reside for ER families. The paragraph added reads:

‘Military families may be a useful basis of comparison to ER families in that they experience a partner who has exposure to trauma and a high tempo job, where occupational and emotional demands may impact on the family (70). Whilst not all aspects of military life may be directly translatable, such as deployments with longer separations, more military families are living dispersed within communities (71) and their experiences may shed light on understanding potential mental health and wellbeing risks in ER families’

Lines 456 – 459.

Another comparison between ER family and military family. I feel military ‘lone parenting’ is the extreme version, as the parent at home is truly alone for however long their partner is deployed. ER families rarely experience this level of lone parenting outside of extreme disasters. Other occupations by which to compare may be more appropriate (Dr? Nurse?)

- We have qualified this statement and also given another example that reads:

‘Other occupations that report non-standard working times, such as evenings, nights and weekends, akin to many ER shift patterns, also find individuals report time-based work family conflict, where the working partner cannot be around to partake in family activities, leaving the other partner with the family responsibilities (73). Although the nature and frequency of familial separation and strains of ongoing micro-transitions seem distinct for families of ER, this description of family life and pressures experienced may be similar in some respects to military spouses/partners who describe lone parenting and episodes of separation and disruption to family life due to deployments (70).’

Lines 486-488

I’m ok with this comparison between ER and military. Perhaps discuss the shift towards more balanced genders in these roles (at least for paramedicine this is true, unsure re other occupations) and whether this might/not change the macho environment etc.

- We have added a sentence to read:

‘It is unknown from the available literature whether ER professions with more of a gender balance and caring vocations such as paramedics may have less aggressive spillover behaviours into the family environment. Therefore in lieu of this evidence, DV/IPV in these professions needs to be a future area of investigation.’

Lines 494 - 496.

I’m ok with this comparison between ER and military. Vicarious trauma is definitely something partners/children, (even researchers and journalists) can experience.

- Thank you

Lines 528-530.

Good. Perhaps discuss the difficulty of confidentiality in these professions. ER (and military) are not at liberty to discuss work activities- this differs from many occupations where you can informally debrief with your loved one after work.

- We have added this point in to read:

‘There may however be extra barriers in ER occupations of communication where they are not at liberty to share details of experiences due to confidentiality. Hence, supporting ER couples to engage in effective communication, where issues and emotions are discussed without breaking confidentiality, could be protective and may be particularly pertinent if the ER partner experiences mental health problems’

Line 543

I think the word ‘non’ is not required?

We have clarified this to read:

‘These findings support one study in this review that found maternal occupational exposure to trauma, in the partner who was not in the ER occupation,’ 

Lines 553 – 556.

I’m ok with this comparison to military too. Highlights research gaps.

- Thank you

Line 601

Suggests the research was conducted after the COVID-19 pandemic. I’d argue we are still within the pandemic? Agreed the impact of the pandemic on ER families must be researched.

- We’ve amended this to read:

‘Whilst this review was conducted during the COVID-19 pandemic,’

Line 618-622

Suggest there are other limitations to this systematic review- data bases searched etc, understand word limitations may have restricted explanation.

- We have added this to limitations

Appendix 3

Love the detail in this table, however it looks more like the Data Extraction Tool than a succinct summary for a reader. If full data is required to be available, maybe suggest this is supplementary material, and a shorter version supplied for the article?

- Thank you – yes we suggest that Appendix 3 is supplementary material for readers if they wish to have in-depth information from the papers and have the underlying data available to access more easily. We have also refined this table and reduced it by 10 pages to make essential information clearer for readers.

In addition, there is some inconsistency with reporting data (if staying with more in-depth table). Consider Stanford (ref 47) and compare to Davidson (ref 34). Davidson has quite an in-depth explanation of measures, while Stanford does not.

- We have reviewed our table and refined sections to aid consistency, presentation and reduced text. 

Also, significant statistics being in bold for Davidson (ref 34) compared to Roberts (ref 48) which is not.

- We have reviewed our table and refined sections to aid consistency, presentation and reduced text.

There is a need for an abbreviation index/key in this appendix.

- We have ensured that in each paper section that abbreviations are spelt out in full once with brackets detailing the abbreviation to be used.

Gibson (ref 52): The strain was largest in BOTH anger and depression?

- We have clarified the reporting of these statistics and included the final logistic regression with all variables included so as not to cause confusion. Strain was largest in model 1 for anger (not including depression in the model) and strain was largest in model 2 for depression (not including anger in the model). 

Haddock (ref 39): Ever divorce? Definition required.

- we have amended the text to read ‘ever having divorced’ – the measurement of this is described in the ‘Measures’ section.

Pfefferbaum (ref 67) Groups A/B/C/D explained in the column after being mentioned previously. Define at first use.

- We have amended this entry

There are a few lines which need proof reading, the one I highlighted: Bochantin (ref 43) ‘This study existing work–family research by moving beyond it being…’.

- We have proofread our table again and corrected mistakes

Richardson (ref 68) Coding revealed 11 themes. Given the extent of your other explanations for other articles, what are those 11 themes?

- We have added in description of these themes

---

## [Editor Report · Decision Letter 1]

26 May 2022

The mental health and wellbeing of spouses, partners and children of emergency responders: a systematic review

PONE-D-21-23204R1

Dear Dr. Sharp,

We’re pleased to inform you that your manuscript has been judged scientifically suitable for publication and will be formally accepted for publication once it meets all outstanding technical requirements.

Kind regards,

Marianna Mazza

Academic Editor

PLOS ONE
---

## [Editor Report · Acceptance letter]

6 Jun 2022

PONE-D-21-23204R1 

The mental health and wellbeing of spouses, partners and children of emergency responders: a systematic review 

Dear Dr. Sharp:

I'm pleased to inform you that your manuscript has been deemed suitable for publication in PLOS ONE. Congratulations! Your manuscript is now with our production department. 

Kind regards, 

on behalf of

Dr. Marianna Mazza 

Academic Editor

PLOS ONE